

# Patterns of telomere length with age in African mole-rats: New insights from quantitative fluorescence in situ hybridisation (qFISH)

Stephanie R.L. Leonida[1,2], Nigel C. Bennett[2], Andrew R. Leitch[1] and Chris G. Faulkes[1]

[1] School of Biological & Chemical Sciences, Queen Mary University of London, London, UK
[2] Department of Zoology, University of Pretoria, Pretoria, South Africa

Corresponding author
Chris G. Faulkes,
c.g.faulkes@qmul.ac.uk

## ABSTRACT

Naked mole-rats *Heterocephalus glaber* (NMRs) are the longest-lived rodent and also resist the normal signs of senescence. In a number of species, cellular ageing has been correlated with a reduction in telomere length, yet relatively little is known about telomeres and their age-related dynamics in NMRs and other African mole-rats. Here, we apply fluorescence in situ hybridisation (FISH) to quantify telomeric repeat sequences in the NMR, the Damaraland mole-rat, *Fukomys damarensis* (DMR) and the Mahali mole-rat, *Cryptomys hottentotus mahali* (MMR). Both terminal and non-terminal telomeric sequences were identified in chromosomes of the NMR and DMR, whilst the MMR displayed only terminal telomeric repeats. Measurements of tooth wear and eruption patterns in wild caught DMRs and MMRs, and known ages in captive bred NMRs, were used to place individuals into relative age classes and compared with a quantitative measure of telomeric fluorescence (as a proxy for telomere size). While NMRs and MMRs failed to show an age-related decline in telomeric fluorescence, the DMR had a significant decrease in fluorescence with age, suggesting a decrease in telomere size in older animals. Our results suggest that among African mole-rats there is variation between species with respect to the role of telomere shortening in ageing, and the replicative theory of cellular senescence.

## INTRODUCTION

In most eukaryotes, telomere length reduction with age is a direct outcome of incomplete replication of chromosome ends during replication; the 'end replication problem' (*Hayflick & Moorhead, 1961*; *Lansdorp et al., 1996*; *Zakian, 2012*). Once telomeres reach a critically short length, cellular signals initiate the process of cell cycle arrest and subsequent senescence, known as the 'replicative theory of ageing/senescence'. Studies of telomere length dynamics in humans and birds have supported this theory, finding a strong correlation between telomere length and lifespan, with a decrease in telomere length with age (*Vleck, Haussmann & Vleck, 2003*; *Harley, Futcher & Greider, 1990*; *Harley, 1995*;

*Haussmann & Vleck, 2002*; *Haussmann, Vleck & Nisbet, 2003*; *Heidinger et al., 2012*). Telomere-dependent replicative senescence is also seen as an anti-cancer mechanism in humans, as progressive telomere shortening ultimately leads to permanent arrest of cell proliferation. While in some species there is good evidence for a link between telomere length and life history, it is less clear whether there is a direct causal role between them.

It is thought that long-lived species have evolved a suppression of telomerase activity to mediate replicative senescence. It is argued that there has been selection in larger animals (which tend to live longer) to have evolved suppressed telomerase activity to alleviate an increased cancer risk arising from their large number of cells and increased lifespan, which provides increased opportunities for cancer-inducing mutations to occur ('Peto's paradox'). This hypothesis has been suggested to apply generally in mammals (*Gomes et al., 2011*), including rodent species (*Seluanov et al., 2007*), where telomerase activity has been shown to correlate with body mass.

Naked mole-rats (NMRs) may be an anomaly to Peto's hypothesis—they are a small rodent with extreme longevity (potentially living up to 37 years), but with high levels of telomerase activity (*Tian et al., 2018*). Alternative telomere-independent mechanisms appear to have evolved in NMR to suppress cancer-inducing mutations (*Gorbunova et al., 2014*; *Tian, Seluanov & Gorbunova, 2017*) and to resist the normal signs of senescence (*Buffenstein, 2005, 2008*) over their lifespan. This article examines NMRs and two other species of African mole-rats in the family Bathyergidae with different longevities to compare age-related dynamics in telomere length: *Fukomys damarensis* (or Damaraland mole-rat, DMR) and *Cryptomys hottentotus mahali* (or Mahali mole-rat, MMR) with estimated lifespans of 11 and 15 years respectively (*Weigl, 2005*). We apply the technique of quantitative peptide nucleic acid-FISH (PNA-QFISH) to (i) identify terminal and/or non-terminal telomeric repeats in the NMR, DMR and MMR, and (ii) quantify changes in telomere size with animal age.

## MATERIALS AND METHODS

### Animals

Animals were caught using Hickman traps baited with sweet potato (*Hickman, 1979*; *Arslan, 2013*). A total of 12 DMRs were used in this study (4 Males and 8 Females), captured on a farm near Tswalu Kalahari Reserve, South Africa (27.2961°S 22.3943°E), with permission from Department of Environment and Nature Conservation, Northern Cape Province (Permit No: FAUNA 171/2/2015). A total of 27 MMRs (13 Males and 14 Females) were caught on private plots in northern Pretoria (25.6567°S 27.9547°E), with permission from Gauteng Department of Nature Conservation (Permit number: CPF6-0127). The study was approved through the Animal Ethics Committee, University of Pretoria, South Africa, AUCC project number: EC037-15. For NMRs, samples from 13 animals (8 Males and 5 Females) were obtained from captive bred colonies kept at Queen Mary University of London. Tissue was obtained from post-mortem specimens from animals free from disease in compliance with national (Home Office) and institutional procedures and guidelines.
Stage of molar tooth wear and eruption pattern was used to place individual DMRs and MMRs into one of five, or one of four age classes in order to give an approximation of age as described previously for DMRs and *Cryptomys hottentotus* (*Bennett, Jarvis & Wallace, 1990*). NMRs of known ages (from birth records) together with the DMRs and MMRs of estimated ages were placed into an age category of young, middle-aged or old to enable a more standardised comparison for further analysis (Table S1). No samples were obtained from middle-aged DMRs.

## Bone marrow extraction and FISH

Bone marrow lymphocyte cells were extracted using a standard protocol (https://www.jax. org/research-and-faculty/tools/cytogenetic-and-down-syndrome-models-resource/ protocols/marrow-preps-protocol) from all 12 DMRs, all 27 MMRs and 9 of the 13 NMRs. Because of their smaller size, four of the NMR samples failed to yield sufficient cells for further analysis. Temperature for incubation of bone marrow cells was modified to mimic the internal body temperature of each mole-rat species; 32 °C (NMR) and 35 °C (MMR and DMR). Cells were prepared and stained with DAPI and Cy3 PNA probe with Telomere PNA FISH Kit (K5326; DAKO, Glostrup, Denmark) using the protocol provided. A mixed population of lymphocyte cells (50 cell replicates for each animal sampled), each being diploid un-replicated cells (2C value) were chosen from each sample preparation for fluorescence analysis.

## Microscope and image capture

Fluorescence in situ hybridisation (FISH) was conducted on interphase cells and representative metaphase spreads and quantitative FISH (qFISH) measurements of total telomere fluorescence of interphase cells from DMR, MMR and NMR were acquired on a Leica DMRA2 upright microscope equipped with a CoolSNAP HQ CCD camera using the 100× 1.4 NA Oil immersion objective, with DAPI and Cy3 filters. The aperture window was set at 1.25× and kept constant during image capture. The exposure time was kept constant for each species (330 ms for DMR and NMR; 470 ms for MMR), when using the Cy3 filter for image capture of telomere spots of interphase cells and metaphase spreads. Telomere fluorescence was measured using OpenLab version 5.5.0.

The mean telomere fluorescence was measured for 50 randomly selected non-overlapping, circular or near circular interphase cells from each individual for each species. Samples from each species were batch processed together to mitigate against inter-assay variation. To verify that cells were in a state where chromosomal material remained unreplicated, measurements of mean DAPI fluorescence were recorded (cells with double the amount of fluorescence would have replicated their chromosomal material). DAPI fluorescence measurements confirmed that cells chosen for telomere fluorescence measurements had not replicated their DNA, that is they were in G1 phase of interphase.

## Calculation of total telomere fluorescence

A background measurement of a blank area (not containing interphase cells and that of a specified area) was taken for each image to be analysed. The area of this measurement was

kept constant for each image captured for consistency across samples. The mean background measurement was then subtracted from the mean fluorescence of telomere spots to produce the total telomere fluorescence 'volume' (intensity × area) expressed in arbitrary units for a given cell.

### Statistical analysis

All analyses were conducted using the R statistical software programme (Version 3.2.5). None of the fluorescence data (for the NMR, DMR and MMR) satisfied the criteria of a normal distribution required for a two-level nested ANOVA. Despite the data being transformed by a number of methods, it remained positively skewed and divergent from normality (Shapiro–Wilk test: $p < 0.0001$). To overcome this and increase robustness, the data were ranked to decrease the within group variance, and total lymphocyte telomere signal fluorescence was tested against age class using a nested ANOVA. Within the nested ANOVA model, the total volume of telomere fluorescence was input as the response variable, with the relative age class and individual identity input as explanatory variables. A Tukey post-hoc test was used to assess the difference in variation between telomere fluorescence of young, middle-aged and old MMRs.

## RESULTS

Terminal telomeric repeats were successfully detected and visualised in the DMR, MMR and NMR (Fig. 1). Two homologous chromosome pairs in the DMR clearly share interstitial telomeric repeat sequences (ITSs) along their chromosome arms, illustrated by sections of diffuse qFISH staining. Additionally, a single homologous chromosome pair appears to have a shorter tract of these repeat sequences (fainter staining area seen in some cells; Fig. 1B). The NMR potentially had ITSs on one chromosome pair (Fig. 1D), although the region of increased fluorescence was faint, but noticeable. In contrast, MMR chromosomes appeared to display a strictly terminal telomeric morphology, with no apparent ITSs observed on any chromosomes (Fig. 1C).

The karyotype for the DMR samples from our collection site ($2n = 78$) was consistent with previous studies (*Nevo et al., 1986*). We confirmed that the MMR had the same chromosome number as other mole-rats in the *Cryptomys hottentotus* clade ($2n = 54$ for females, and 53 for males; Fig. S1) including *Cryptomys h. hottentotus*, *Cryptomys h. natalensis* and *Cryptomys h. pretoriae* (*Nevo et al., 1986*; *Faulkes et al., 2004*; *Deuve et al., 2008*).

In DMRs, although there was a significant difference in total fluorescence among individuals within age classes ($F_{24,1371} = 27.05$, $p < 0.001$), overall there was also a significant reduction in total telomere fluorescence in the bone marrow cells of old DMRs compared with young animals ($F_{2,1371} = 10.14$, $p < 0.001$; Fig. 2A). For the MMR, significant variation in telomere fluorescence of bone marrow cells was found between individual MMRs within each group ($F_{24,1371} = 27.05$, $p < 0.001$), and among groups of young, middle-aged and old individuals ($F_{2,1371} = 10.14$, $p < 0.001$; Fig. 2B). A post-hoc Tukey test revealed a small, but significant increase in telomere fluorescence in old vs middle-aged and young MMRs ($p < 0.001$ and $P < 0.05$ respectively; Fig. 2B). Finally, as

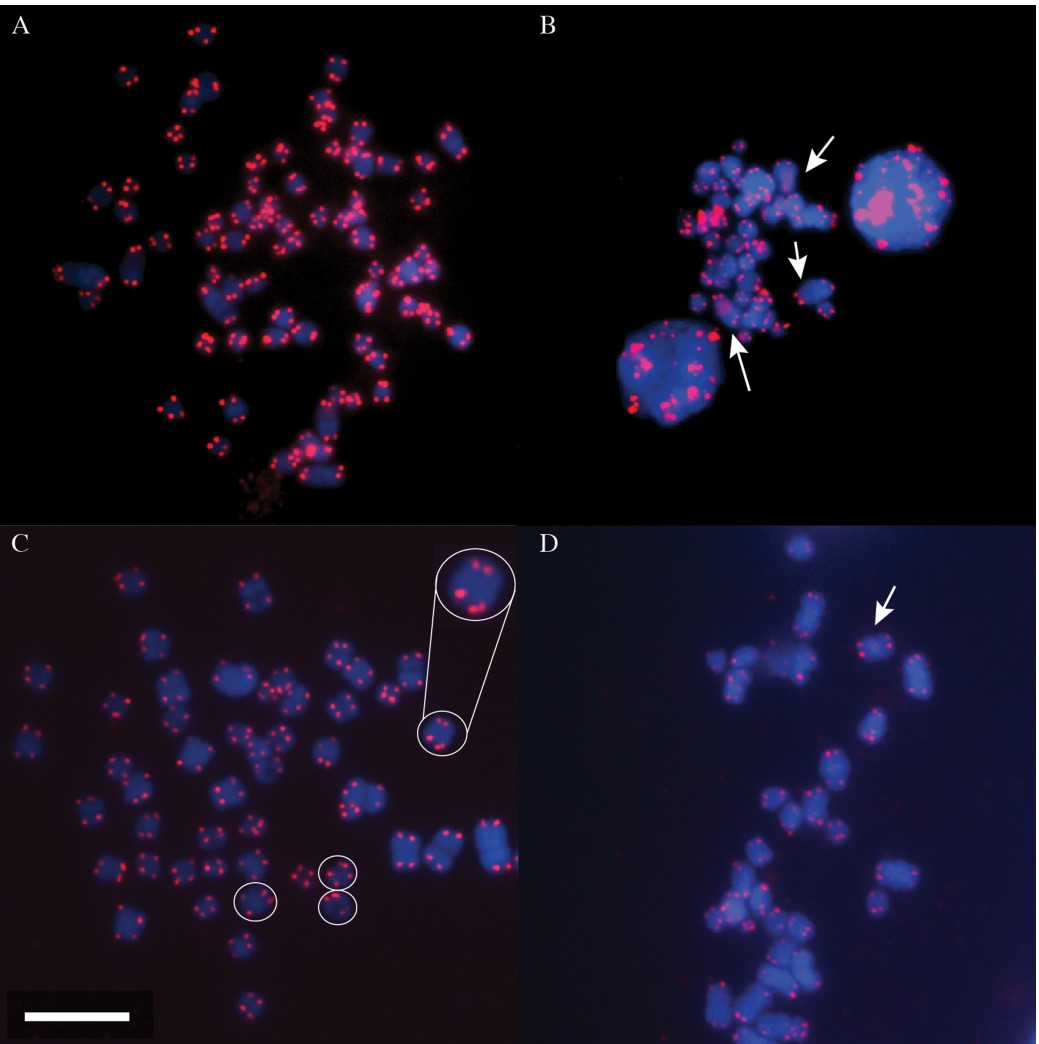

**Figure 1 Metaphase spreads of DMR, MMR and NMR chromosomes.** DNA is stained in blue (DAPI) and telomeres in red (Cy3 PNA probe). (A) Metaphase spread of DMR chromosomes. (B) Partial metaphase spread of DMR chromosomes showing diffuse interstitial telomeric sequences (ITSs, arrowed, with middle arrow showing a very weak signal seen in some cells; see text). Note that some chromosomes are partially overlapping. (C) Partial metaphase spread of MMR chromosomes. Note how some chromatids have double signals at the telomere, presumably representing differential condensation of the telomeric array (circles and expanded inset for one chromosome pair). A full image of this spread is given in Fig. S1. (D) Partial metaphase spread of NMR chromosomes. White arrow indicates a faint region of increased fluorescence that may be a potential region of ITSs. White scale bar = 10 μm; all images at the same magnification.

with the other two species, NMRs had significant differences in telomere fluorescence within age classes ($F_{6,441} = 12.01$, $p < 0.001$), and between classes ($F_{2,441} = 23.90$, $p < 0.001$). Total telomere fluorescence did not decline with age—old vs young animals were not significantly different ($p = 0.21$), while middle aged animals had a significant increase in fluorescence compared to young and old age classes ($p < 0.001$). Individual animal sample means with standard errors plotted against age in years are shown on Fig. S2.

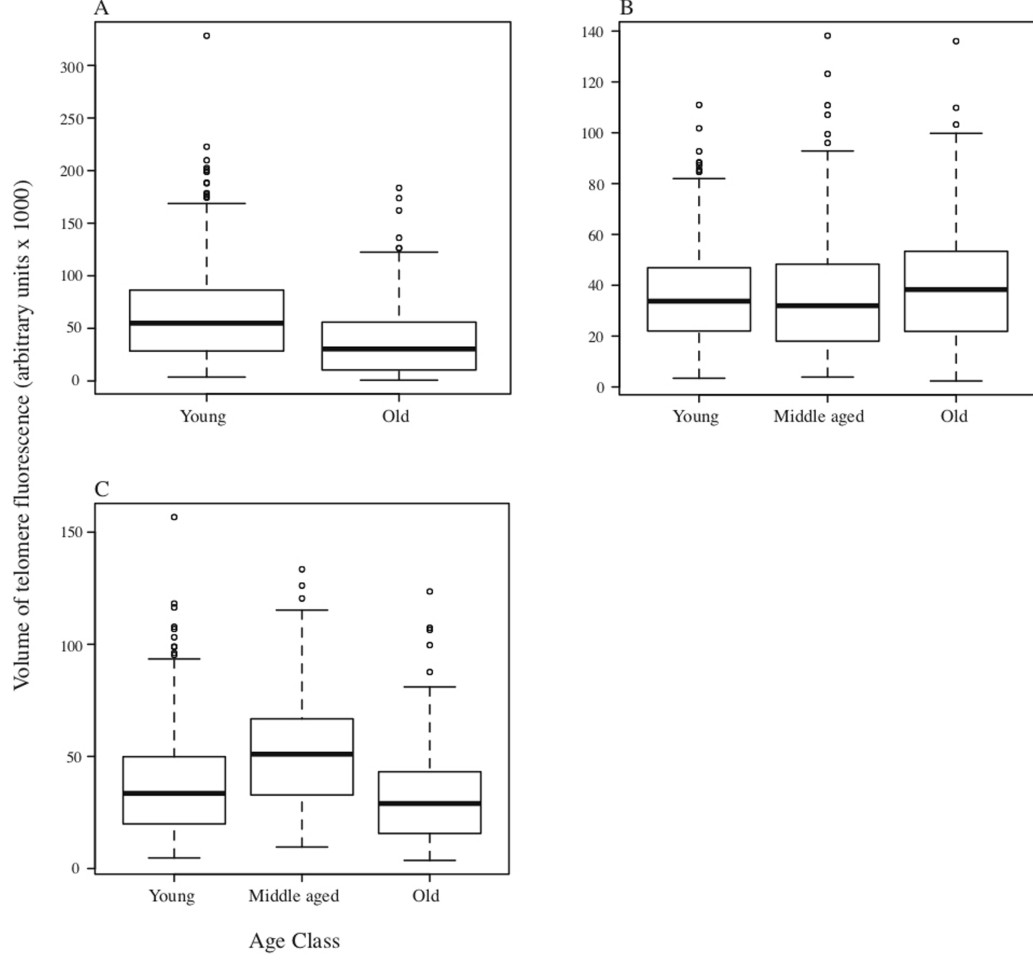

**Figure 2 Boxplots depicting medians and interquartile ranges for telomere size (expressed as volume of telomere fluorescence) variation with age class.** (A) DMR ($n$ = 6 animals in each age class); (B) MMR ($n$ = 8, 14 and 6 individuals for young, middle-aged and old age classes respectively); (C) NMR ($n$ = 3, 4 and 3 individuals respectively).

## DISCUSSION

The use of qFISH for calculating telomere length is well established as a research tool (e.g. in mouse strains: *Zijlmans et al., 1997*), comparable with telomere length measurement method using Southern blotting/in situ hybridisation techniques and in some applications may be more precise than qPCR methods (*Hultdin et al., 1998*; *Gutierrez-Rodrigues et al., 2014*). Apart from mammals, qFISH has also been employed as a single technique estimator of telomere length in diverse vertebrate taxa such as fish (*Panasiak, Dobosz & Ocalewicz, 2020*) and reptiles (*Olsson et al., 2020*). PNA-qFISH clearly identified terminal telomeric repeats in the DMR, MMR and NMR and we report for the first time ITSs in the DMR and tentatively in the NMR (Figs. 1A and 1D). Chromosomes containing ITSs as observed in the DMR and potentially the NMR may indicate active karyotype evolution, and potentially mark regions of fusion of ancestral chromosomes or sites of double-strand break repair in unstable areas of the genome

(*Meyne et al., 1990*; *Azzalin, Nergadze & Giulotto, 2001*; *Ruiz-Herrera et al., 2008*). Such molecular instability can lead to further breakage or fission and chromosome fusions. It is well known that the DMR karyotype is among one of the most rearranged karyotypes amongst members of the family Bathyergidae (*Deuve et al., 2008*). Variation in chromosome number to produce chromosomal races within the DMR has resulted in $2n$ varying from 74 to 78 (*Nevo et al., 1986*) and also $2n = 80$ individuals can occur in different geographic areas (*Deuve et al., 2008*). This chromosomal variation, coupled with the observation of ITSs, suggest that the DMR genome has ongoing, active karyotype evolution, in contrast to the fixed karyotypes seen in the MMR and others in the *Cryptomys hottentotus* clade (*Deuve et al., 2008*). Future work could involve using G and C-banding techniques and chromosome painting using whole chromosome paints to observe association between heterochromatic regions and ITS and terminal telomeric repeats (*Yang et al., 2004*). This may allow more detailed insights into the various types of past chromosomal rearrangements that have occurred in the DMR and other species in the genus *Fukomys*, where there is extensive karyotypic variation among species (*Van Daele et al., 2004*).

The presence of double telomere signals at the end of some chromatids in the MMR (Fig. 1C) is a feature that has been reported previously in other mammals and may arise through telomere—nuclear envelope interactions. Such interactions are proposed to protect against recombination of telomeric repeats during mitosis and are considered essential for correct meiotic pairing of homologous chromosomes during meiosis (*Schober et al., 2009*; *Chikashige et al., 2006*; *Ding et al., 2007*).

In all three species we found significant intra-specific differences in total telomere fluorescence among individuals. Variance in telomere size among individuals has been widely reported in other species, and some studies have shown associations between early-life telomere length and individual life history for example European badgers *Meles meles* (*Van Lieshout et al., 2019*), and meerkats (*Cram et al., 2017*). The significance of individual variation in telomere sizes in mole-rats identified in our study cannot be addressed further without additional longitudinal studies using PCR-based techniques to quantify telomere size. We also found interesting species differences in the variation in telomere fluorescence with age, although a limitation of the study was being able to accurately age the wild caught DMRs and MMRs. Our results revealed a significant reduction in telomere fluorescence and by implication a reduction in telomere size with age class when comparing young and old DMRs (Fig. 2A), suggesting that telomere shortening is a cause or biomarker for cellular senescence in this species. In contrast, both MMRs and NMRs did not show an age-related decline in total telomere fluorescence (Figs. 2B and 2C). Moreover, in both species we observed an increase in telomere fluorescence with age compared to the young age category, although our sample size was small for the NMR. These observations do not support the hypothesis of cellular senescence in these species and support other indirect evidence for telomere maintenance with age published for the NMR. *Seluanov et al. (2007)* found that telomerase activity in NMRs was high, and comparable with other small (but short-lived) rodents like

mice and gerbils that have a high incidence of cancer. Furthermore *Kim et al. (2011)* showed stable levels of TERT expression (that functions to maintain telomeres) regardless of age (comparing a new-born, 4 and 20 year old NMRs). Recently, *Adwan Shekhidem et al. (2019)* used a different (qPCR-based) approach to show that telomere length in the NMR did not shorten with age, instead showing a mild elongation. The telomeric fluorescence signal in the NMR here suggest elevated activity of telomere elongating processes in middle aged animals. Conversely, *Adwan Shekhidem et al. (2019)* found that telomere length in the cancer resistant, long-lived blind mole-rat *Spalax* declined with age, in a similar fashion to short-lived rodents, similar to our observations here for DMR. Collectively, these results highlight some fascinating variation in telomere dynamics within the African mole-rats, and among other rodents, revealing the potential for evolutionary gains and losses of mechanisms associated with telomere maintenance, cancer resistance, longevity and senescence.

# ACKNOWLEDGEMENTS

Daniel Hart, Ruth Rose and Elizabeth Archer are thanked for their support and guidance in the lab. We thank Steve Le Comber, Sally Faulkner and Joanne Littlefair for advice and help with the statistical analysis.

## Funding

This work was supported by the SARChi Chair for Mammalian Behavioural Ecology and Physiology GUN 64756 to Nigel C. Bennett. The funders had no role in study design, data collection and analysis, decision to publish, or preparation of the manuscript.

## Grant Disclosures

The following grant information was disclosed by the authors:
Mammalian Behavioural Ecology and Physiology: GUN 64756.

## Competing Interests

The authors declare that they have no competing interests.

## Author Contributions

- Stephanie R.L. Leonida conceived and designed the experiments, performed the experiments, analysed the data, prepared figures and/or tables, authored or reviewed drafts of the paper, and approved the final draft.
- Nigel C. Bennett conceived and designed the experiments, authored or reviewed drafts of the paper, and approved the final draft.
- Andrew R. Leitch conceived and designed the experiments, authored or reviewed drafts of the paper, and approved the final draft.
- Chris G. Faulkes conceived and designed the experiments, performed the experiments, analysed the data, prepared figures and/or tables, authored or reviewed drafts of the paper, and approved the final draft.

## Animal Ethics

The following information was supplied relating to ethical approvals (i.e., approving body and any reference numbers):

Ethical considerations for this project were approved through the Animal Ethics Committee, University of Pretoria, South Africa, project number: EC037-15.

Because tissue sample collection was post-euthanasia, in full accordance with National (Schedule 1 of the Animals (Scientific Procedures) Act 1986) and Institutional animal care and use guidelines, additional local ethical approval for naked mole-rat work was not required for this study.

## Field Study Permissions

The following information was supplied relating to field study approvals (i.e., approving body and any reference numbers):

Permission to collect C. h. mahali was granted by the Gauteng Department of Nature Conservation authorities [C. h. mahali: Permit No CPF6-0127] and for F. damarensis permission was granted by the Department of Environment and Nature Conservation, Northern cape Province [F. damarensis: Permit No FAUNA 171/2/2015].

## Data Availability

The raw measurements of fluorescence are available in the Supplemental files.

## Supplemental Information

Supplemental information for this article can be found online at http://dx.doi.org/10.7717/peerj.10498#supplemental-information.

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
