# Peer review of "Patterns of telomere length with age in African mole-rats: New insights from quantitative fluorescence in situ hybridisation (qFISH)"

_PeerJ, doi:10.7717/peerj.10498_

## Round 0.1 · original submission · Minor Revisions

Authors should take into account the comments of the reviewers on preparing the manuscript in accordance with these comments.

·

Basic reporting

Figure 1D: It is very hard to see the interstitial telomere signal. Please provide another example.

A more detailed description of the FISH TL method used will be helpful to critically review the results. For example, how were the samples batched? Was there a quality control sample? Were there technical replicates? If so, what was the assay coefficient variation of the replicates? What criteria were used to choose the 50 cells analyzed in each animal?

Experimental design

Molar tooth wear and eruption patterns were used to place animals into different age groups for DMR and MMR because these animals were caught in wild and there was no other way of knowing their age. But NMR is bred in-house with age records. So, it would make sense to analyze TL with known age in the NMR group. Relatedly, as the exact age of DMR and MMR is not known, the authors should discuss this as a limitation of this study.


An independent telomere length method, such as qPCR, should be considered to verify the FISH results. It is understandable that the quantity of specimens is challenging especially with bone marrow samples. It will be helpful even if it is done in only a subset of samples.

Validity of the findings

It is possible that the lack of age correlation in NMR and MMR is caused by the high intra-specific differences in the telomere fluorescence signals and small sample size. This point needs to be discussed.

Figure 1: It appears that the telomere signals vary significantly among DMR, MMR, and NMR. Can you comment on whether this reflected different telomere lengths in these three species or technical variations?

Additional comments

Overall, the manuscript provided new sight on the telomere length in these three African mole-rats.
Addressing the small sample issue and the method details will be extremely helpful to bolster the conclusions.

Reviewer 2 ·

Basic reporting

no comment

Experimental design

Calculation of total telomere fluorescence in qFISH experiment for telomere length estimation should cite a proper published research paper.

Validity of the findings

Telomere length measurement by qFISH only might not be enough to support the conclusions of the manuscript. Therefore, telomere length comparisons between individuals of the same species with 3 different ages must be performed using the “gold standard” telomere length measurement method—Southern blotting with a radioactive 32P or DIG-or Biotin-labelled TG-rich probe. Alternatively, could be more challenging, the PCR-based method to measure the telomere length difference between different age groups, consistent results will be more supportive.

·

Basic reporting

no comment

Experimental design

no comment

Validity of the findings

no comment

Additional comments

The study by Leonida et al analyze telomere dynamics with age in three species on mole rats. They show that telomeres shorten in DMRs, but don not change or slightly lengthen in MMR and NMR (the longest-lived species). The study provides novel information on diversity and evolution of telomere maintenance mechanisms in these unique species of rodents.
The study is clearly written and a pleasure to read. I enthusiastically recommend publication.

Minor suggestions:
1. NMR shows dimmer telomere fluorescence compared to DMR. This is consistent with the earlier observations that NMRs have human-sized telomeres (10-12 kb), while other short-lived rodents tend to have longer telomeres (30+ kb). The authors may want to comment on it.
2. Figure 2B has chromosomes that appear not separated from each other. A better image or an explanation in the legend would be helpful.

Vera Gorbunova

---

## Round 0.2 · Major Revisions

I ask the authors to prepare reasoned comments on comments of reviewer #2, and, if possible, to complete the manuscript with the necessary data.

Reviewer 2 ·

Basic reporting

no comment

Experimental design

no comment

Validity of the findings

I am not convinced to support the publication of this manuscript. At least another independent telomere length method must be performed to obtain consistent experimental results as shown in this version. Current data are basically one-dimensional, which can lead to ambigous conclusions.

·

Basic reporting

no comment

Experimental design

no comment

Validity of the findings

no comment

Additional comments

The authors have addressed my concerns

---

## Round 0.3 · accepted · Accept

The manuscript is ready for publication, all comments from the reviewers have been addressed.